# Attachment towards the Owner Is Associated with Spontaneous Sleep EEG Parameters in Family Dogs

**DOI:** 10.3390/ani12070895

**Published:** 2022-03-31

**Authors:** Cecília Carreiro, Vivien Reicher, Anna Kis, Márta Gácsi

**Affiliations:** 1Doctoral School of Biology, Institute of Biology, ELTE Eötvös Loránd University, 1117 Budapest, Hungary; vivien.reicher@gmail.com; 2Department of Ethology, Institute of Biology, ELTE Eötvös Loránd University, 1117 Budapest, Hungary; vargane.kis.anna@ttk.mta.hu (A.K.); marta.gacsi@gmail.com (M.G.); 3MTA-ELTE Comparative Ethology Research Group, 1117 Budapest, Hungary; 4Institute of Cognitive Neuroscience and Psychology, Research Centre for Natural Sciences, 1117 Budapest, Hungary

**Keywords:** affective neuroscience, social bond, behavior, dog–human interaction, emotion regulation, canine EEG, sleep architecture, brain activity, animal model

## Abstract

**Simple Summary:**

Dogs have been shown to form attachment bonds towards their owners analogous to the human infant-parent attachment. In humans, the neurological background of variation in attachment and similar trait-like social behaviors has been described. It is known that certain sleep parameters are in association with an individual’s attachment-related traits. In the current study, we provide the first evidence that dogs’ attachment towards their owner is also associated to dogs’ sleep structure (the time they spend in the different sleep stages) as well as to their brain activity during sleep. Thus, as in humans, when dogs sleep in a novel environment (in the presence of their owners), differences in their attachment bond are reflected in their sleep EEG characteristics.

**Abstract:**

Affective neuroscience studies have demonstrated the impact of social interactions on sleep quality. In humans, trait-like social behaviors, such as attachment, are related to sleep brain activity patterns. Our aim was to investigate associations between companion dogs’ spontaneous brain activity during sleep (in the presence of the owner) and their relevant behavior in a task-free social context assessing their attachment towards the owner. In random order, each dog participated in a non-invasive sleep electroencephalogram (EEG) measurement and in the Strange Situation Test (SST) to assess their attachment behavior. We found that higher attachment scores were associated with more time spent in NREM sleep, lower NREM alpha power activity and lower NREM alpha–delta anticorrelation. Our results reveal that, when dogs sleep in a novel environment in the company of their owners, differences in their attachment are reflected in their sleep EEG characteristics. This could be best explained by the different degree that owners could be used as a safe haven in an unfamiliar environment and during the unusual procedure of the first EEG measurement.

## 1. Introduction

Attachment is behaviorally defined as a form of social bond, in which the attachment figure (e.g., mother) represents a secure base for the attached individual (e.g., children) [1]. In the case of separation or danger, the individual seeks for a ‘safe haven’ to buffer the stress, displaying proximity-seeking behaviors [2]. The neural system related to attachment is typically activated under stress (e.g., separation, facing strangers) when the dependent individual avoids negative stimuli and seeks safety by the proximity of the attachment figure [3]. Attachment studies in humans have demonstrated its impact on social bonds [4], personality traits [5], depression and anxiety disorders [6]. Further, different attachment styles can be related to subjective (e.g., questionnaire) sleep characteristics, for instance, individuals classified with secure attachment style sleep better than those classified with insecure attachment style [7].

Sleep quality, assessed also by objective measures such as electroencephalogram (EEG), was found to be in association with attachment (e.g., [8]). Women with high attachment anxiety, who sleep without a regular partner compared to married women, have shorter deep sleep duration (non-rapid eye movements; NREM stage 3–4) [9]. However, children separated from the mother have a longer deep sleep as a sign of stress [10]. Further, positive associations between attachment and NREM alpha activity (which is an assumed marker of hyperarousal) is also observed in patients with sleep disorders [8]. In non-human species, it is observed that social isolation decreases slow-wave activity in NREM (e.g., delta band in mice) [11], indicating difficulty to down-regulate their hypervigilant state even during sleep. Additionally, the delta band shows an inverted correlation with alpha band associated with emotion processing (e.g., anxiety trait) [12]. The so-called alpha–delta anticorrelation has been observed in anxious individuals in isolation. This anticorrelation reflects a mutual inhibitory interaction between the vigilance-related alpha system, where the alpha band promotes avoidance behaviors, and the reward-related delta system, where the delta band regulates approach behaviors [13].

In general, sleep studies are performed in the unfamiliar environment of laboratories. In order to have a more naturalistic sleep, different designs have been tested and significant differences have been found compared to the sleep research at laboratories. In a study with humans analyzing the relationship between sleep and bedpartners [14], young and elderly couples slept together in their homes. In these familiar surroundings, the results were the opposite of what is usually found in laboratories. In their homes, young men slept better than young women, who had more deep sleep than at a laboratory. Additionally, no difference between genders was found in elderly couples. Another study analyzed how experimental manipulation affects typical home sleeping arrangement in sleep research [15]. It was found that, in the home sleeping together condition, individuals had less deep sleep (stage 4) than individuals who slept alone at the laboratory. A study with dogs also investigated how the environment affects their sleep (e.g., location, time of the day) [16] and results similar to humans were found (e.g., more REM sleep in dogs sleeping at home vs. at the lab).

Besides the aforementioned attachment studies with laboratory animal models (e.g., [11]), most recently, the dog became one of the most promising new model species in neurobiology, in some respects allowing for more valid cross-species comparisons than in rodents and primates [17,18]. Dogs and humans have shared similar physical and social environments for thousands of years, developing a series of convergent behavioral and socio-cognitive features [19]. Behavioral studies provide evidence that, compared to wolves, dogs developed unique competences in social interactions with humans (e.g., [20]), including attachment. Though in non-human animals attachment bond typically develops between the offspring and the mother (e.g., guinea pig [21]; squirrel monkey [22]), dogs develop individual attachment relationships towards their owners [23]. Results from several studies support that even adult family dogs can develop interspecific attachment behaviors towards their owners [24], which is functionally analogous to the human infant-mother relationship (for review, see [25]), including the two major features of the attachment figure (owner): the secure-base effect [26] and the safe-haven effect [27].

So far, fully non-invasive neurocognitive assessments have been performed only in a few species (e.g., cows [28]; horses [29]). However, in the case of family dogs, the use of non-invasive neural methods is more comparable to the procedures applied in human participants (e.g., [18,30,31]). Recent research using non-invasive neuroimaging (i.e., fMRI) has shed some light on the neural background of dogs’ social bonds to their owners (e.g., [32]). Similarly to humans, attachment to the caregiver proved to show a significant association with the reward system in dogs; their owners’ neutral voice evoked higher activation of the reward system in dogs with higher attachment scores [33].

Regarding sleep research specifically, ample evidence shows similarities between human and dog sleep patterns (for review, see [34]). For instance, having a negative socio-emotional experience (e.g., separation from the owner and threatening approach of a stranger) before the sleep session resulted in a redistribution of time spent in the different sleep stages (e.g., decreased durations of drowsiness and NREM) compared to the sleep session after positive experiences (e.g., petting) [35]. In that study, individual differences in dogs’ reaction to the negative (vs. positive) emotional treatment, in terms of sleep pattern changes, showed association with their specific behavioral responses during the pretreatment (e.g., dogs that spent more time standing next to the door during the separation episode had an even shorter duration of NREM after the negative treatment compared to less behaviorally reactive dogs).

Additionally, in dogs, similarly to humans [36], the sleep pattern can also be affected by several factors such as age [30,37,38]. In the case of juvenile dogs, age was positively associated with drowsiness duration (2–8 months old). Further, age (8–14 months old) was negatively associated with delta and positively with alpha power activities [37]. In the case of adult dogs (2–8 years old), NREM delta activity showed a negative association, while NREM alpha activity showed a positive association with age [30].

In this study, we focused on neural correlates of behavioral patterns of attachment in a task-free social context. In randomized order, dogs participated in a daytime sleep measurement and the behavioral test. During the sleep measurement, dogs slept spontaneously in the presence of the owner. To assess attachment, we used the adapted version of the Strange Situation Test (SST) [39]. Based on the findings described above in humans and mice, we assumed that dogs’ attachment towards their owners would be related to the durations of drowsiness and NREM as well as to NREM alpha band and alpha–delta anticorrelation.

## 2. Materials and Methods

The locations of the assessments were a behavioral laboratory equipped with cameras to record the test and a fully equipped laboratory for canine EEG measurements at the neurolaboratory of Eötvös Loránd University or the Research Center for Natural Sciences, Institute of Cognitive Neuroscience and Psychology, depending on the laboratory availability.

### 2.1. Subjects

Owners were recruited from the Family Dog Project (Eötvös Loránd University, Department of Ethology) database. The criteria to include dogs were a minimum age limit of 2 months old with no upper limit and that dogs would be living with the owner for, at least, 2 weeks. For more details of dogs’ demographic data, see Appendix A. A total of 43 dogs in the presence of their owners were assessed in random order for the SST and the sleep EEG on different days. The time interval between the tests did not exceed the 25% of the dogs’ age and no order effect of the tests was found (Appendix A). One dog was excluded because the total EEG record was shorter than 1 h. Thus, we analyzed the data of 42 family dogs (M*_age_* ± SD: 2.29 ± 2.78 years old; age range: 2.5–117.9 months old; 14 males and 28 females) of various breeds (34 purebred dogs from 16 breeds and 8 mixed breed dogs). Dogs varied in size (weight range: 5–35 kg) and had various types of training status.

### 2.2. Behavioral Assessment

To assess dogs’ attachment towards their owners, we tested them in the SST that was originally developed to observe the infant-mother relationship [40] and later adapted for investigating the dog-owner bond [23]. The main principle of SST is the secure-base effect that the attachment figure (owner) exerts on the attached individual (dog). The dog is exposed to moderate stress, which is caused by the unfamiliar place, separation/reunion from the owner and the presence of a stranger. The stress elicited by this test situation activates the attachment system, resulting in observable behavioral characteristics. Based on these behavioral variables, three factor scores can be calculated: attachment to the owner, anxiety in a strange place and acceptance of an unfamiliar person [23,39,41]. In this study, we focused on the attachment score to assess the associations between the dog-owner bond and dogs’ sleep EEG. For the description of the SST behavioral variables related to dogs’ attachment towards the owner, see Table 1 (see details in [39]).

### 2.3. Sleep EEG Recording

All subjects in the presence of their owners were assessed in a non-invasive sleep EEG recording with minimum duration of 1 h and maximum of 3 h during daytime. This variance in recording time occurred due to differences in subject compliance. When some dogs woke up, they often became active and, in cases of such awakening, the recording had to be finished regardless of the length of elapsed time. Some dogs were able to fall asleep again even after short awakenings and the recording continued until the maximum duration of 3 h.

The sleep occasion occurred in a relatively active day [37,42]. Before the measurements, the experimenter (E) explained the procedure to the owner while the dog was allowed to explore the room for 5 to 10 min. After that, the owner settled on a mattress with the dog and helped E by gently holding the dog while the electrodes were placed on the surface of the dog’s head. During the electrode placement, all dogs were rewarded with food and/or social reinforcement (e.g., petting, praise). In cases where the dog did not appear calm and/or comfortable with the electrode placement, E added brief breaks (~5 min). In the case of 8 young dogs, the electrode placement could not be finished within 1 h, so the owner was asked to come back with the dog a second time. Of those invited back for a second attempt, all of them were successfully measured. After the electrode placement and checking the appropriate quality of the EEG signals, owners were asked to mute their cell phones and engage in a quiet activity such as reading, watching a movie on an electronic device with earphones or sleeping. E left the room and monitored the measurement on the computer next to the sleep laboratory. In the case of the rare event of malfunction of an electrode, E re-entered the laboratory and replaced or changed the electrode.

Dogs were measured between 2015–2020. During this period, our laboratory improved and updated the electrode placement and recording methods. In the case of 17 dogs, the previous setup was used, thus, one active EEG channel (frontal: Fz) and an eye movement channel were recorded. In the case of 25 dogs, the current setup was used, specifically four active EEG channels and an eye movement channel were recorded. Fz along with Cz channel were placed over the anteroposterior midline of the skull; F7 and F8 channels were placed on the right and left zygomatic arch next to the eyes, all referred to G2. The ground electrode (G1) was attached to the left *musculus temporalis*. Furthermore, an additional channel, labeled EOG, was visualized as the bipolarly referenced F7-F8 electrodes. In all dogs, at least the frontal electrode (Fz) was active, thus, data from this electrode were used for spectral analyses.

For electrode placement, SignaSpray Electrode Solution (Parker Laboratories, Fairfield, NJ, USA) was used to separate dogs’ fur where gold-coated Ag/AgCl electrodes were attached onto the skin using EC2 Grass Electrode Cream (Grass Technologies, West Warwick, RI, USA). Impedance values of the EEG electrodes were kept under 20 kΩ during the recordings. For visualization of electrode placement, see Figure 1.

Recordings were obtained with one of the following technical arrangements:(1)In the case of 25 dogs (59.5% of the total sample), the signal was collected, amplified and digitized at a sampling rate of 1000 Hz/channel using the 40-channels NuAmps amplifier (© 2018 Compumedics Neuroscan) and DC-recording, later saved in .cnt format with the Scan 4.3 Acquire software (© 2018 Compumedics Neuroscan) then converted to .edf format using MatLab EEG Toolbox.(2)In the case of 17 dogs (40.5% of the total sample), the signal was collected, pre-filtered, amplified and digitized with a sampling rate of 1024 Hz/channel using a SAM 25 R style MicroMed Headbox (MicroMed Inc., Houston, TX, USA). The passband was set at 0.5–256 Hz, using System Plus Evolution software (MicroMed Inc, Houston, TX, USA), which exported data in .edf format.

To correct for differences in EEG filter characteristics across recording devices, a standard calibration process (dog [37,38]; human [43,44]) was implemented on devices (1) and (2). Specifically, a waveform generator at the Fz electrode input of both devices was used to apply 40 and 355 μV amplitude sinusoid signals at various amplitudes (0.05 Hz, every 0.1 Hz between 0.1–2 Hz, every 1 Hz between 2–20 Hz, every 10 Hz between 10 Hz–100 Hz). The amplitude reduction rate for each recording system was determined by calculating the proportion of digital (measured) and analog (generated) amplitudes of sinusoid signals. Next, amplitude reduction rates were calculated for each device and EEG spectrum amplitudes were corrected by dividing such calculated values by the obtained amplitude reduction rate for the recording system.

### 2.4. Data Analysis

Sleep recordings were visually scored in accordance with standard criteria [45] adapted for dogs [30]. A program developed by Ferenc Gombos (Fercio’s EEG Plus, 2009–2022) was used to analyze and export the data. The recordings were manually scored identifying the stages of wake, drowsiness, NREM and REM [46], artifacts on the EEG channels were excluded and the program provided data of macrostructural and spectrum variables from the different sleep stages. Regarding the macrostructural variables, we focused on drowsiness and NREM durations. Despite the fact that, e.g., sleep latency is also reported as an important factor in human studies, this variable had a large variation in our sample due to its very strong association with age (possibly mediated by subject compliance and, thus, time needed for the experimenter to fix the electrodes). Furthermore, although REM sleep plays an important role in emotional processing, it was not included because our study did not involve any handling to directly evoke emotions; rather, we assessed attachment, which is a relatively stable trait with not much evidence that it is related to REM. Thus, we focused on the other sleep stages as suggested by some authors studying attachment in animal models (e.g., [7]).

Relative EEG power spectra were calculated only for NREM sleep as this is the stage where all dogs had sufficiently long (>6 min) artifact-free traces (criteria based on [37]). The macrostructural and the spectrum variables of interest are described in Table 2. Due to the variance of total record time (M*_record_duration_* ± SD: 168.59 ± 32.81 min), we calculated the relative durations related to each subject’s record duration.

### 2.5. Statistical Analysis

The statistical analyses were performed in R (version 4.1.2: R Core Team, 2021). Based on Shapiro–Wilk normality test, SST attachment score and the sleep EEG macrostructural variables (relative drowsiness and NREM durations) showed normal distribution. Given the non-normal distribution of the EEG spectral variables, NREM relative delta and alpha bands were normalized using natural logarithmic transformations [47] and all variables of interest were treated by means of parametric tests.

As all EEG variables correlated with age (that showed non-normal distribution), in the final analysis, Kendall’s partial rank correlations were performed to determine the relationship between attachment and sleep EEG variables whilst controlling for age. To control for multiple comparisons, Benjamini-Hochberg correction was conducted and results based on the adjusted *p*-values are reported.

## 3. Results

### 3.1. Sleep Macrostructure

Attachment had a positive association with relative NREM sleep duration (*rp* = 0.263; *ps* = 0.008; Figure 2) and it was not associated with relative drowsiness duration (*rp* = −0.124; *ps* = 0.254; Figure 3). Actual power analysis showed that NREM sleep had a small effect size (Cohen’s d = 0.243; provided by G*Power version 3.1.9.7) [48].

### 3.2. Spectral Analysis (in NREM Sleep)

The negative association of attachment with alpha power (*rp* = −0.231; *ps* = 0.025; Figure 4) and alpha–delta anticorrelation (*rp* = −0.225; *ps* = 0.044) was significant even after controlling for age. Actual power analysis showed that alpha power (Cohen’s d = 0.634) and alpha–delta anticorrelation (Cohen’s d = 0.650) had a medium effect size (provided by G*Power version 3.1.9.7) [48].

## 4. Discussion

In our study, we measured the first sleep occasion of dogs without specific treatment prior to the EEG, with positive reinforcement during the electrode placement (e.g., petting, motherese, food) on a relatively active day (i.e., not highly loaded active day). Though the first sleep occasion has been shown to be affected by the sleep habits of dogs (i.e., first-night-effect-like effects) [42], it is also proven to be a valid marker of trait-like characteristics. Overall, the first sleep occasion is a situation stressful enough to affect the sleep parameters and potentially evoke the attachment system in dogs. Our results supported the broad hypothesis; namely the behavioral dimension of attachment, assessed by SST, showed associations with the spontaneous brain activity, measured by non-invasive EEG.

Regarding sleep macrostructure, dogs with higher attachment scores spent more time in NREM sleep, while no association with drowsiness sleep was found. This result is somewhat paralleled by previous human data, e.g., showing that married women with high anxious attachment who sleep in the company of a regular partner spend more time in deep sleep [9].

In a previous EEG study on family dogs, participation in a negative social interaction, which included separation from the owner, induced a decrease in NREM duration during post-interaction sleep [35] compared to a positive treatment. The difference in the dogs’ pre-sleep stress was claimed to be the major effect that resulted in the change of the sleep macrostructure. Our results are rather in line with the data of the positive pretreatment group, which showed a longer time spent in NREM [35], suggesting that dogs with higher attachment scores might have had a more stable inner state during the measurement due to the secure base/safe haven providing role of their owner.

Of note, our results cannot be directly compared to the findings of other experiments as, in most studies of attachment-related behaviors, the setup involves some negative experience, such as isolation (e.g., children, mice) [10,11] and social conflict (e.g., rats) [49]. Thus, individuals in negative conditions show an increase not just in NREM duration, but also in intensity (mainly delta power), together reflecting a stronger need for restorative sleep. In our study, dogs with higher attachment scores showed an increased NREM duration; but no increased NREM intensity was observed (indicated by the alpha–delta anticorrelation), suggesting a lower need for restorative sleep when sleeping in the presence of the owner.

Dogs’ higher attachment scores were associated with lower NREM alpha power. A higher alpha activity, a marker of hyperarousal, was found in human patients with sleep disorders, when they were sleeping alone [8]. In dogs, sleeping in the presence of the owner, we found a lower alpha activity associated with higher attachment scores. Alpha is related to cortical activity, thus, it is typically dominant in wakefulness and associated with perceptual awareness, vigilant state and inhibitory control. Alpha interference during sleep is usually associated with stressful stimulation, anxiety trait and the persistence of a general vigilance state [12,50]. When thalamocortical circuits are stimulated, increased alpha reflects the effort to suppress this potential sleep disturbance [51]. Insecurely attached individuals were reported to have a higher alpha activity in unfamiliar places and alone [52,53], indicating their unmet need for proximity [54]. Verbeke et al. [55] concluded that anxiously attached participants, seated in silence in the presence of a stranger with no interaction, failed to have their need for approval met and became preoccupied, which was reflected by their higher alpha power. In our study, although dogs slept in an unfamiliar place, they had the proximity of their attachment figure, their owners. Dogs with higher attachment scores had lower NREM alpha power, indicating the safe-haven effect of the owner [27] to buffer the stress of sleeping in an unfamiliar place and the handling for the EEG.

Furthermore, dogs’ higher attachment scores showed a lower alpha–delta anticorrelation. In humans, studies of emotion processing show a strong alpha–delta anticorrelation in individuals with high anxiety [12] and depression scores [56]. These findings indicate that stronger negative feedback from the alpha to delta system is a characteristic feature of individuals with emotion processing deficit and a stronger inhibitory control executed by vigilance-related alpha system over reward-related delta system [13].

A vast number of sleep studies focuses on the longer duration of prior wakefulness and its influence on NREM sleep, i.e., sleep pressure and homeostasis. Thus, as part of our EEG protocol, owners were instructed to keep the dog in a relatively active day, e.g., no extra activity carried out or sleep deprivation, which would increase sleep pressure and affect NREM variable as shown in previous sleep studies in dogs [16,30]. Furthermore, some studies examining the influence of the nature of what animals experience before sleeping reported results similar to ours [49,57,58]. However, we did not have systematic information about either the owner compliance or on the exact duration of time spent awake and this specific control (e.g., groups of different hours dogs are awake before the sleep EEG) would be valid for further studies.

Due to practical reasons, our sample size presented a marked age variance, thus, we considered possible age effects on the EEG variables. Although previous findings demonstrated EEG alterations along the dog brain maturation [37] and age [30], which were also confirmed by our analysis, in the current study, age had little influence on the associations between the behavioral and the EEG parameters. Yet variations of attachment at different ages may have influenced the small-to-medium effect size observed. Future studies, however, should use a sample with a more homogeneous age and also control for dogs’ sleep experience in novel environments [42] to decrease potential confounding factors. Studies including dogs younger than 8 months of age should not be considered as adult samples when analyzing drowsiness duration due to early brain development similar to children [37]. Regarding NREM power spectra, it has been observed that the dog central nervous system is not fully mature by 12 months of age, as they do not stabilize even by the age of 14 months (e.g., [37]). Furthermore, future analyses might be favored by the use of other biomarkers, such as heart and respiratory rates [59], allowing to reveal subtle physiological responses related to differences in the attachment bond [60]. Additionally, an analysis comparing conditions of sleeping with the owner vs. alone (as in humans) [55], might vastly improve the knowledge about how similar family dogs are in this respect to humans. Finally, owner’s characteristics can also influence dog–owner interactions and should be considered in further studies of dogs’ sleep quality. For instance, polymorphisms in the owners’ OXTR gene are related to their dogs’ attachment behavior in the SST test [41], whereas the owners’ personality and interaction style influences their dogs’ reaction to stressful situations (e.g., threatening approach) [61] or when fulfilling simple commands [62].

## 5. Conclusions

To our knowledge, the current study is the first to examine, in a task-free social context, the associations between behaviorally measured dog–owner attachment and sleep EEG macrostructure as well as EEG spectrum. A higher attachment score of dogs to their owners was linked to higher NREM duration and lower NREM alpha power and alpha–delta ratio. Thus, dogs’ attachment score was associated with sleep variables related to emotion regulation, which can be explained by the different degree the owners could be used as a safe haven in the moderately stressful first EEG measurement. These results strengthen the feasibility of the dog as a model for further comparative studies.

## Figures and Tables

**Figure 1 animals-12-00895-f001:**
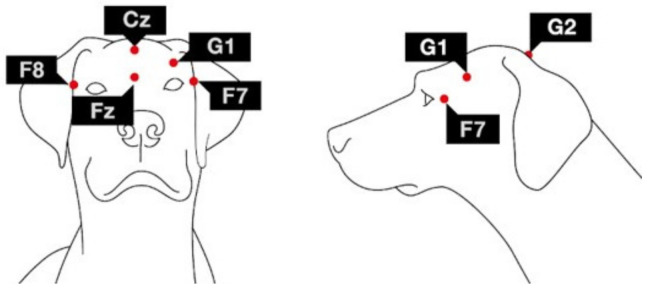
Placement of the electrodes. Fz and Cz: frontal and central midline; F7 and F8: left and right electrodes placed on the zygomatic arch; G2: reference electrode; G1: ground electrode.

**Figure 2 animals-12-00895-f002:**
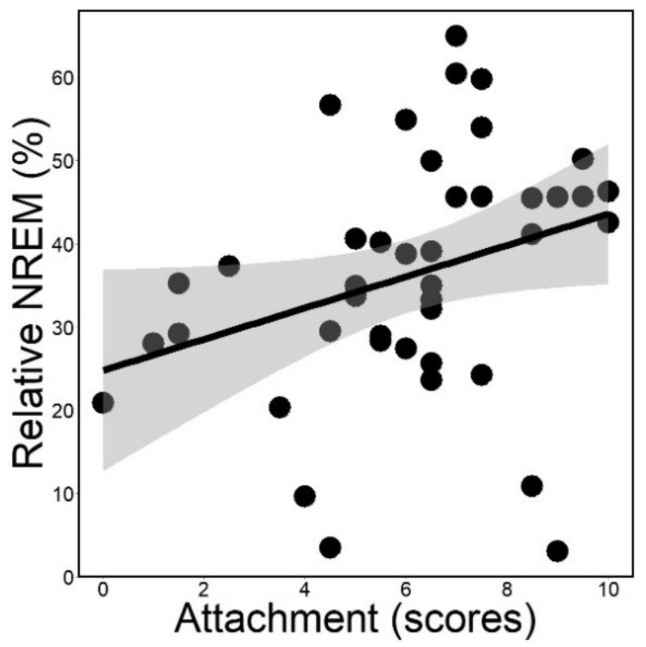
Relative duration of NREM plotted as a function of attachment scores measured in the Strange Situation Test. Gray area: 95% confidence interval from standard error.

**Figure 3 animals-12-00895-f003:**
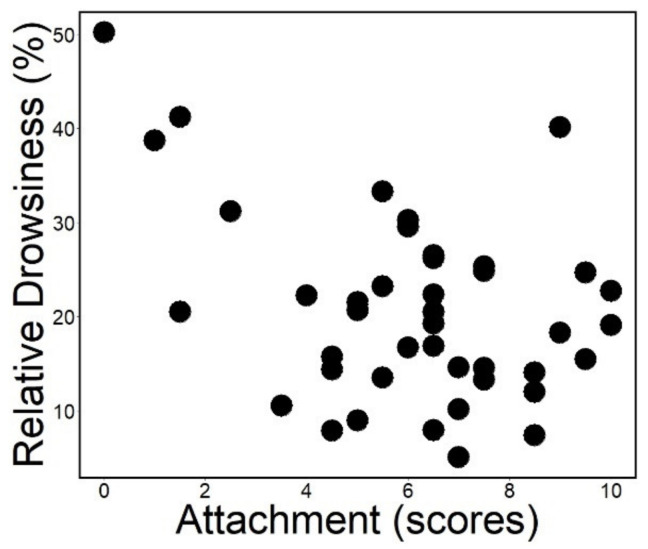
Relative duration of drowsiness plotted as a function of attachment scores measured in the Strange Situation Test. (The association is not significant after controlling for age).

**Figure 4 animals-12-00895-f004:**
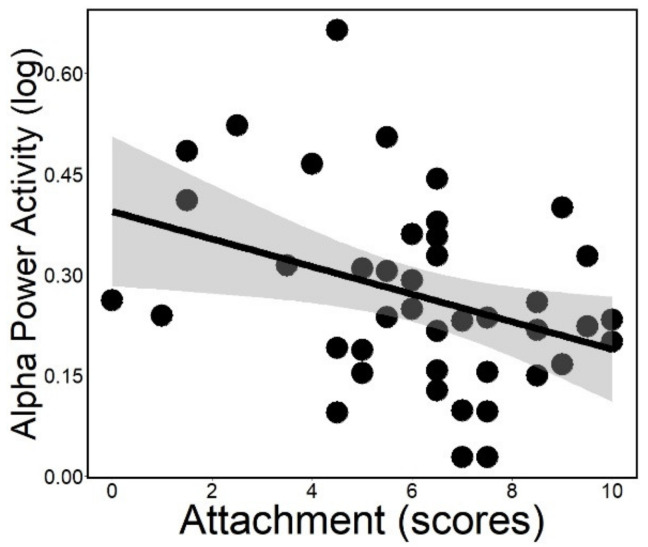
Alpha power activity plotted as a function of attachment scores measured in the Strange Situation Test. Gray area: 95% confidence interval from standard error.

**Table 1 animals-12-00895-t001:** Description of the Strange Situation Test behavioral variables related to dogs’ attachment towards the owner. D: dog; O: owner; m: meter; s: second; S: stranger.

Episode	Variable	Behavior Description	Score
**1, 2, 4, 6**	**Proximity**	D is close to O (closest body part is within 1 m)—in more than 75% of the time when D is not exploring or playing	1
**1**	**BlockO-1**	During the first block-carrying episode, D watches or follows O for more than half of the time	1
**2**	**LeaveO-1**	When O first leaves, D follows O to the door (at least within 1 m from the door)	1
**4**	**EnterO-1**	When O first enters, D approaches O at once (in reaching distance) and wags the tail	1
**4**	**BlockO-2**	During the second block-carrying episode, D watches or follows O for more than half of the time	1
**4**	**LeaveO-2**	When O leaves the second time, D follows O to the door (at least within 1 m from the door)	0.5
**6**	**EnterO-2**	When O enters the second time, D approaches O at once (in reaching distance) and wags the tail/jumps/spins	0.5
**3**	**DoorS-1**	D stands by or orients at O’s door (for at least 5 s—score 0.5; almost all the time—score 1) during first separation	1
**3**	**NoPlayS**	D does not play with S, although D played with S for more than 10 s in Episode 2 (in O’s presence)	1
**3, 5**	**VocalizeS**	D vocalizes (any occurrence, except asking S for ball)	0.5
**3, 5**	**Chair**	D is mostly (for more than half of the time) at O’s chair if not at the door	0.5
**5**	**DoorS-2**	D stands by or orients at O’s door (for at least 5 s) during second separation	1
**5**	**EscapeS**	When S enters, D at first tries to approach the door opening (to sneak out through the door) instead of greeting S	0.5
**6**	**DoorS-3**	D stands by or orients at O’s door (for at least 5 s) during third separation	0.5
		**sum**	**11**

**Table 2 animals-12-00895-t002:** Variables of interest measured by the behavioral test and the sleep measurement.

Assessment	Variable	Measure
**SST ^1^ factor**	Attachment	sum of the behavior items referring to dog’s attachment towards the owner (score range from 0–11)
**Sleep macrostructure**	Relative drowsiness	time spent in drowsiness/record duration (%)
	Relative NREM ^2^	time spent in NREM/record duration (%)
**NREM spectrum**	Alpha band	relative power of the frequency range 8–12 Hz ^3^ (log ^4^)
	Alpha–delta anticorrelation	ratio of relative powers of alpha (8–12 Hz) and delta (1–4 Hz) (log)

^1^ Strange Situation Test; ^2^ Non-Rapid Eye Movement sleep stage; ^3^ Hertz; ^4^ Logarithm.

## Data Availability

The data presented in this study are available on request from the corresponding author.

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
