# Peer review of "Attachment towards the Owner Is Associated with Spontaneous Sleep EEG Parameters in Family Dogs"

_animals, 2022, doi:10.3390/ani12070895_

Round 1
Reviewer 1 Report
This is a well-written and nicely presented study. My main request is that the authors more fully present their results with regard to SST results/scores and age, and whether there were any differences in dogs who had the SST on the same day versus on a separate day from their EEG. The authors suggest in several places that the EEG was performed on a relatively stress-free day, however it appears that the SST would be stressful for some dogs. It is difficult to fully appreciate the implications of the results without this information and analysis of these features. Otherwise I have no major revisions to request - the paper was overall very clear and well-detailed.
Reviewer 2 Report
Brief summary: The paper entitled ‘Attachment towards the owner is associated with spontaneous sleep EEG parameters in family dogs’ focuses on the impact of social interactions on sleep physiology in domestic dogs. The test design comprises 43 dogs with their owners to complete the Strange Situation Test (SST, adapted for dogs). After that, all dogs (n= 43) had the opportunity to sleep between 1 to maximum 3 hours. Sleep was recorded non-invasively with EEG electrodes placed at Fz, Cz positions and on the zygomatic arch (F7, F8) for EOG, referenced against G2. Sleep stages were manually scored for ‘wake’, ‘drowsiness’, NREM and REM sleep and relative EEG power spectra analysis was calculated at Fz for NREM sleep only. Power spectra were then correlated (Kendall’s rank correlations) with the results of the SST. Results show that higher attachment scores were associated with more time spent in NREM sleep, lower NREM alpha power and lower NREM alpha/delta anti-correlation. The authors conclude that both, sleeping in unfamiliar surroundings and lower attachment score, are associated with less restorative sleep characterised by higher alpha- (as a sign of arousals) and less delta-power.
General comments:
Study design: The study focuses on the dogs’ attachment towards their owner (evaluated by the Strange Situation Test) without considering the interaction style of the dog owners. There are several studies (e.g. Cimarelli et al., 2017), demonstrating a strong emotional effect of the owners approach to his/her pet (and probably also on sleep), especially in unfamiliar or threatening situations. This shortcoming of the current investigation, which may influence the results, should be mentioned as a study limitation (at the end of the ‘Discussion’).
REM sleep hypothesis of emotional-memory processing: There is significant behavioural and neurophysiological evidence that in humans REM-sleep plays an important role in emotional processing (e.g. review by Tempesta et al., 2018). In the ‘Introduction’, mainly studies with patients are cited (e.g. Muris et al., 2001; Troxel et al. 2007; Sloan et al. 2007) for illustrating the association between alpha activity in NREM, emotions and attachment style. As a matter of facts mood disorders such as anxiety, depression may affect sleep in many ways and thus may not serve here as a valid framework to explain the relationship between attachment style and sleep physiology in domestic dogs. From this perspective, I wonder why the authors did not consider/include REM-sleep in their analyses, since the relationship between emotions and REM-sleep characteristics is well documented (in humans without sleep disorders or other pathologies)!
Sleep pressure and homeostasis: According to current models of sleep physiology and homeostasis, the amount of delta sleep depends of the wake-time prior to sleep episodes (see. Borbély et al., 1982, 1984, 2009). Therefore, it is crucial to assess the time awake prior to any sleep study, especially with focus on NREM (delta) sleep. In this context, wake periods of one or two hours longer may already cause higher sleep pressure and - consequently – also more sleep. In the current investigation, information about pre-sleep wake periods are missing, which might be crucial in the interpretation of the reported findings.
Presence/absence of bed/pet-partner: The study by Troxel et al. (2007) mentioned in the ‘Introduction’ focused on women with recurrent major depression - probably sleeping alone in the sleep lab – but in the absence of their (bed)-partners. Although Troxel et al. reported less deep sleep in this cohort of anxious and depressed patients, the effects of bedpartners (presence/absence) on sleep was not investigated. Although the influence of environmental factors on sleep-physiology in humans is not fully understood, there are several studies suggesting that the social setting (familiar environment and the presence/absence of bedpartners, children) are relevant for sleep quantity and -quality.
Since the authors also wanted to investigate on ‘dogs sleep in a novel environment in the company of their owners’ (line 26, 27), the paper by Troxel et al. is not enough. Additional literature is necessary to explain the influence of partner on sleep. For example:
- Monroe (1969) reported significant differences in stage 4 sleep (minutes) and REM sleep (minutes) with less deep sleep and more REM sleep in the sleeping together condition;
- Butte et al. (2015) investigated in young and elderly couples with less deep sleep and REM-sleep in females but not in older couples when sleeping at home e.g. in familiar surroundings.
Spectral EEG analysis: Spectral EEG-analyses was performed on Fz electrodes only. Iotchev et al. (2018) investigated on canine sleep and reported sleep spindles, especially slow spindles (8 to 13 Hz) predominantly seen at the Fz electrode position. Since slow spindles may overlap with the alpha frequency band (8 to 14 Hz), the authors should clarify if (and in which way) this was controlled for in their quantitative EEG analysis.
Specific comments, suggestion:
Line 454, literature 42. Hobson (1969) reviewed in the article the scoring manual by Rechtschaffen & Kales. Since this is not the original scoring manual, please correct the literature to ‘A. Rechtschaffen & A. Kales (Eds.): A manual of standardized terminology, techniques and scoring system for sleep stages of human subjects. Public Health Service, U.S. Government Printing Office, Washington, D.C., 1968’
By the way: It is not clear why the authors refer to the scoring rules by Rechtschaffen & Kales (R&K, 1967) since in 2007 the R&K-criteria were replaced by new rules of the American Academy of Sleep Medicine (Iber et al. 2007). The AASM-criteria offer additional rules, which might be helpful in canine sleep scoring (e.g. merging deep sleep stages 3 and 4 into N3, alternative electrode placements and additional EEG traces, rules for scoring arousals, etc.).
Reviewer 3 Report
This ms reports results of a non-invasive EEG study with dogs, and correlates with their responses to an adapted strange situation test, a well-established method of determining dog-owner attachment.
This is an interesting report and it will make a good contribution to the literature. I just have a few suggestions for clarification on a couple of things.
L47 - the brackets here are confusing. It makes it unclear what the sentence is trying to say. Suggest rewording and removing the brackets, if possible.
Subjects - having a table with the demographic info about each dog would be helpful here, including the training level, age, and breed.
L160 - were owners permitted to touch the dog at all during the EEG recording? Were there any restrictions on the dogs' ability to lay in certain positions, etc? All of these things could impact the quality of the EEG recording.
Procedure - the authors note that the electrode placement evolved over the course of the data collection period, and apparently at some point mid-data collection, the headbox was replaced. This is OK, but I'd like to see some evidence that neither of these things impacted the results.
L230 - 'all EEG variables correlated with age...' This implies that preliminary analyses were done to figure out whether any demographic variables impacted the results. However, these are not reported anywhere. Please describe these analyses and present the results somewhere. Similarly, were there any differences depending on the order of SST/EEG recording?
Figs 2 and 4 - please describe the shaded area in the caption. What does it represent? SE? SD?
Discussion - in addition to the age-related issues noted as a limitation in the discussion, it also be worth discussing the fact that all sig results had a small effect size. This should also ideally be noted in the results as well.
